# HiFi metagenomic sequencing enables assembly of accurate and complete genomes from human gut microbiota

Chan Yeong Kim[1,3,4], Junyeong Ma [1,4] & Insuk Lee [1,2] ✉

Advances in metagenomic assembly have led to the discovery of genomes belonging to uncultured microorganisms. Metagenome-assembled genomes (MAGs) often suffer from fragmentation and chimerism. Recently, 20 complete MAGs (cMAGs) have been assembled from Oxford Nanopore long-read sequencing of 13 human fecal samples, but with low nucleotide accuracy. Here, we report 102 cMAGs obtained by Pacific Biosciences (PacBio) high-accuracy long-read (HiFi) metagenomic sequencing of five human fecal samples, whose initial circular contigs were selected for complete prokaryotic genomes using our bioinformatics workflow. Nucleotide accuracy of the final cMAGs was as high as that of Illumina sequencing. The cMAGs could exceed 6 Mbp and included complete genomes of diverse taxa, including entirely uncultured RF39 and TANB77 orders. Moreover, cMAGs revealed that regions hard to assemble by short-read sequencing comprised mostly genomic islands and rRNAs. HiFi metagenomic sequencing will facilitate cataloging accurate and complete genomes from complex microbial communities, including uncultured species.

Despite advances in culturomics techniques, most human gut prokaryotic species remain uncultured[1–3]. Therefore, conventional cataloging of microbial genomes, based on the isolation of clonal genomic DNA followed by sequencing and assembly, may not be applicable to all human gut commensals. De novo metagenomic assembly using human fecal sequencing samples has proven useful in reconstructing the genomes of gut species including uncultured taxa[2,3]. Nevertheless, these metagenome-assembled genomes (MAGs) are generally discontinuous because of conserved, repetitive, and mobile sequences[4,5]. Long-read metagenomic sequencing using Oxford nanopore technology (ONT) with short-read error correction has enabled the assembly of 20 circularized complete MAGs (cMAGs) from 13 human stool samples, albeit with low nucleotide accuracy[6]. PacBio SMRT long-read sequencing with ultra-deep short-read sequencing also assembled four cMAGs from 12 human fecal samples, but their nucleotide accuracy

was not reported[7]. More recently, PacBio high-accuracy long-read (HiFi) sequencing, which has become popular for the assembly of reference animal and plant genomes[8,9] has been applied for the analysis of complex microbiomes, such as sheep fecal samples[10] and chicken cecum samples[11]. HiFi repetitive sequencing of a circularized SMRTbell library calls reads by consensus, substantially improving nucleotide accuracy while maintaining long read length. Moreover, specialized assemblers for HiFi metagenomic assembly, hifiasm-meta[11] and metaFlye[12], enable the highly accurate reconstruction of cMAGs.

In the present study, we conducted an exhaustive assembly of HiFi metagenomic sequencing reads from five human fecal samples. Seeking only circularized contigs, we skipped the binning procedure. We developed a bioinformatics workflow to filter initially assembled circular contigs for complete prokaryotic genomes. Eventually, we obtained 102 cMAGs and investigated their nucleotide accuracy,

[1]Department of Biotechnology, College of Life Science and Biotechnology, Yonsei University, Seoul 03722, Republic of Korea. [2]POSTECH Biotech Center, Pohang University of Science and Technology (POSTECH), Pohang 37673, Republic of Korea. [3]Present address: Structural and Computational Biology Unit, European Molecular Biology Laboratory, Heidelberg, Germany. [4]These authors contributed equally: Chan Yeong Kim, Junyeong Ma. ✉e-mail: insuklee@yonsei.ac.kr

taxonomic diversity, and ability in large genome assembly. We also examined how HiFi metagenome sequencing can complement short-read metagenome sequencing for the study of uncultured species in the microbial communities.

## Results

### HiFi metagenomic sequencing assembles cMAGs belonging to diverse gut microbiota taxa

We obtained public HiFi metagenomic sequencing data from four human fecal samples. Two samples were pooled from vegan donors and the other two from omnivore donors (15.2–18.5 Gb in sequencing depth). In addition, we produced in-house HiFi metagenomic sequencing data based on a fecal sample from a healthy Korean omnivore donor using the Sequel II platform (29.6 Gb in sequencing depth). Compared to a recently published ONT metagenomic sequencing dataset on human fecal samples[6], the HiFi metagenomic sequencing data used in this study displayed similar total base pairs but much longer reads and significantly higher base quality (Methods, Supplementary Fig. 1, Supplementary Data 1).

To obtain as many circular contigs as possible, we assembled HiFi sequencing reads for each sample using three different metagenomic assemblers, metaFlye[12], HiCanu[13], and hifiasm_meta[11], which yielded 2,283, 481, and 590 circular contigs, respectively, or 3,354 in total. Because these circular contigs might contain viral genomes, plasmids, and misassembled closed contigs, we developed a bioinformatics workflow to select only complete prokaryotic genomes (Methods, Fig. 1a). First, we filtered circular contigs using biological priors and structure parameters, including sequence length ≥ 100 kbp, ≥100 genome taxonomy database (GTDB) marker proteins, presence of rRNAs, ≥ 20 tRNA types, and no assembly bubble or repeat (Supplementary Fig. 2a, b). Although metaFlye initially assembled the largest number of circular contigs, ~97.7% (2231) were filtered out using these criteria. Accordingly, 145, 76, and 52 circular contigs passed the first filtering step by hifiasm_meta, HiCanu, and metaFlye, respectively (Supplementary Fig. 2c). Because we applied multiple assemblers on pooled fecal samples, any resulting redundant contigs with near-identical sequences (average nucleotide identity, ANI > 0.99 and maximum alignment coverage > 0.95) were removed after calculating all pairwise genome similarities (Supplementary Fig. 2d–g, Supplementary Data 2).

Repeated sequences could cause early assembly closing, generating prokaryotic genomes with significant gaps[14]. These defective genomes may not be detected even by alignment with conspecific genomes, unless genomes derived from isolates are available for the species. Therefore, we designed another filtering step based on the congruency of each circular contig with its conspecific genomes from a comprehensive catalog of human gut microbial genomes, HRGM (human reference gut microbiome)[3], which includes both MAGs and genomes derived from isolates. We hypothesized that although individual HRGM genomes might not be complete, their core contigs shared among most conspecific genomes likely originated from genuine species. Therefore, we developed a software, cMAGfilter[15], which filters the faulty circular contig based on its congruency with conspecific MAGs. For each circular contig, we first identified conspecific HRGM genomes and determined their core contigs. Then, we aligned the core contigs to the query circular contig, and excluded the latter if the retrieval rate of core contigs was <95% or there were fewer than five conspecific genomes (Supplementary Fig. 3a, b, Supplementary Data 3). Circular contigs that did not meet these criteria were significantly shorter than their conspecific genomes derived from isolates (Supplementary Fig. 3c) and contained significant gaps compared to their closest genome derived from isolates (Supplementary Fig. 3d–f). Overall, our bioinformatics workflow retained 102 of the 3,354 circular contigs initially assembled.

Previously, a skewed GC content was proposed as a metric for verifying the correctness of assembled prokaryotic genomes[16]. We manually examined the cumulative GC-skew pattern of the 102 circular contigs and classified them into five classes: very clear, clear, decent, poor, and no pattern (Supplementary Data 4). Many circular contigs with poor or no GC symmetry patterns possessed the characteristics of genomes from specific lineages rather than misassembled ones. For example, five out of six Oscillospiraceae family genomes showed poor or no GC-skew pattern, and all Gemmiger genus, Bifidobacterium adolescentis, and Bifidobacterium longum genomes showed no GC symmetry (Supplementary Fig. 4). Indeed, the latter two were previously reported as not having GC symmetry[16]. Accordingly, we decided not to filter out circular contigs based on the GC-skew pattern.

The final 102 cMAGs could represent the complete genomes of human gut prokaryotic species. Notably, hifiasm_meta contributed most of the 102 cMAGs (~88.2%) (Fig. 1b). We found that 29 of 102 cMAGs were exclusively assembled from the newly sequenced Korean fecal sample (KR001) (Fig. 1c). Although bias toward a few abundant bacterial clades (e.g., Oscillospirales) was observed, cMAGs from HiFi metagenome sequencing reads covered diverse phylogenetic groups of human gut microbiota. Based on the GTDB annotation, we obtained cMAGs for both Archaea and Bacteria, comprising 9 phyla, 11 classes, 14 orders, 24 families, 52 genera, and 84 species. Among them, 1 phylum, 1 class, 1 order, 4 families, 12 genera, and 22 species are contributed solely by the KR001 sample (Fig. 1d, e, Supplementary Data 5–6).

### The cMAGs from HiFi metagenomic sequencing reads are accurate and can be large

Next, we evaluated the nucleotide accuracy of cMAGs obtained by HiFi metagenomic sequencing. Even without taking into account sequencing errors, same-species prokaryotic genomes assembled from different samples are hardly identical to each other because of genetic variation accrued via horizontal gene transfer and spontaneous mutation. To assess the nucleotide accuracy of cMAGs, we chose Bifidobacterium animalis, which showed significantly lower genetic variation than other gut commensal species. B. animalis (HRGM_Genome_1769) displayed the tenth lowest single nucleotide variation (SNV) rate among the 1,521 HRGM species with known SNV values and the lowest SNV rate among HRGM species with genomes derived from isolates[3] (Fig. 2a). We then aligned the B. animalis HiFi cMAG (OMN01_MFL_0491) to its closest genome derived from isolates in RefSeq[17] (GCF_000224965.2). Interestingly, the two genomes were nearly identical (ANI > 0.9999 and alignment coverage > 0.9999) (Fig. 2b), even though they were assembled at different locations and time points (USA in 2021 and Italy in 2013, respectively). Given that the RefSeq genome (GCF_000224965.2) is for a probiotic bacterium, B. animalis subsp. lactis BLC1[18], the cMAG (OMN01_MFL_0491) may also be derived from probiotics. We next calculated the SNV density of B. animalis HiFi cMAG (OMN01_MFL_0491) against its conspecific HRGM representative genome (HRGM_Genome_1769), and compared it with the SNV density distribution of conspecific genomes (genomes within the species bin of HRGM_Genome_1769) (Methods, Supplementary Fig. 5). We did not observe a substantial deviation in the SNV rate (maximum 14.6%) (Fig. 2c), suggesting that SNVs were attributable mostly to spontaneous mutations and errors occurring during short-read sequencing. To generalize these findings, we explored the percentile rank of the SNV rate among 77 HiFi cMAGs in their conspecific genomes. The 50th percentile rank fell within the interquartile range of the boxplot, and only four cMAGs were above the 99th percentile rank (Fig. 2d). Hence, SNV rates of most HiFi cMAGs do not deviate from the distribution of their conspecific HRGM genomes.

We next examined the size of genomes assembled by HiFi metagenomic sequencing. Four cMAGs in the Bacteroides genus were over 6 Mbp and the longest seven cMAGs were assembled from the Korean sample (KR001), which had the greatest sequencing depth (Fig. 2e,

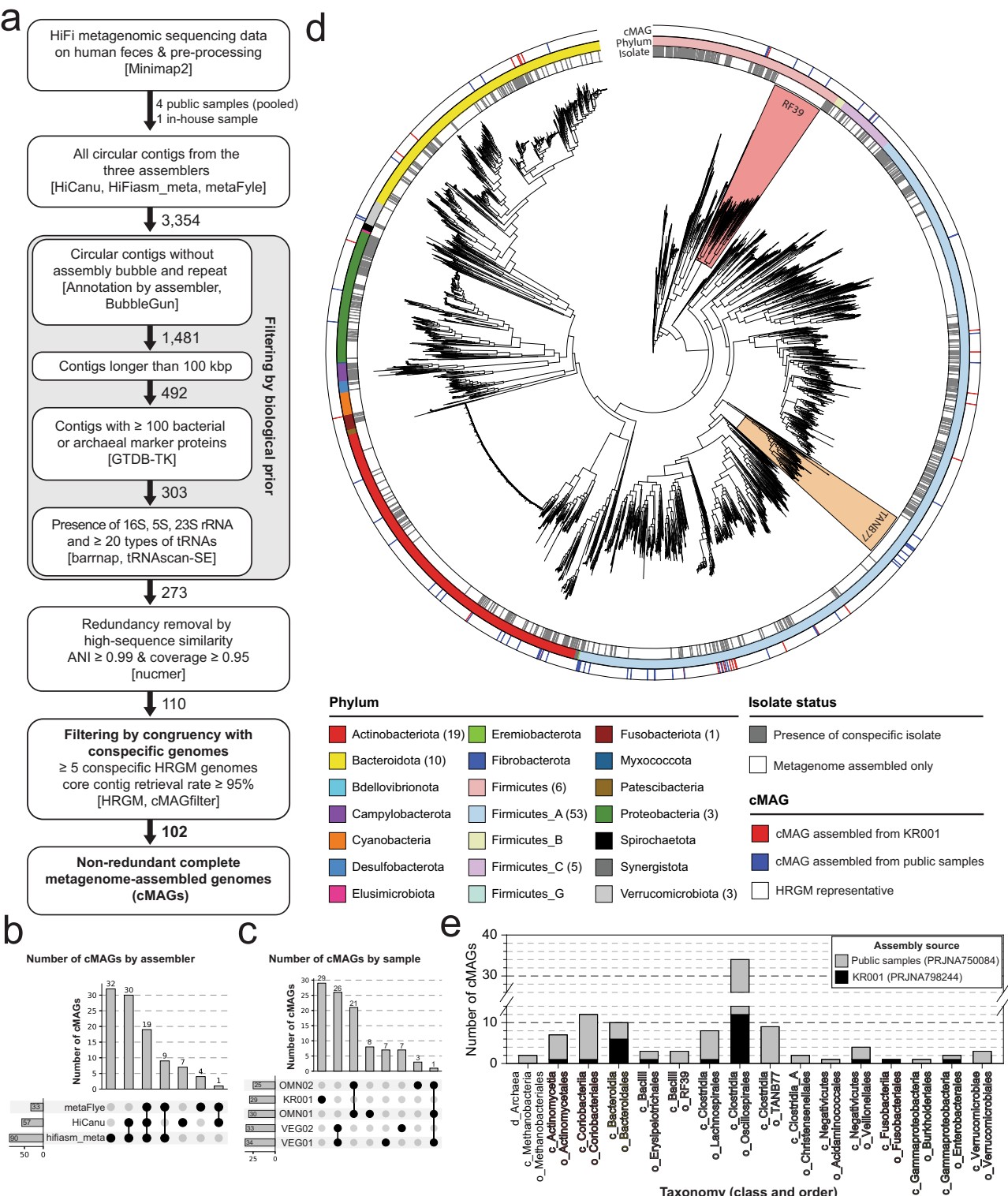

**Fig. 1 | Assembly of 102 complete metagenome-assembled genomes (cMAGs) from human fecal high-accuracy long-read (HiFi) metagenomic sequencing samples. a** Schematic diagram illustrating the HiFi cMAG assembly workflow. **b** Number of final cMAGs assembled from each assembler. Overlapping areas represent redundant genomes assembled by more than one assembler. **c** Number of cMAGs assembled from each sample. OMN and VEG represent the public pooled samples derived from the omnivores and vegans, respectively, and KR001 represent the newly sequenced Korean sample. **d** Maximum

likelihood phylogenetic tree of 5486 bacteria from the human reference gut microbiome (HRGM) and HiFi-assembled cMAGs. RF39 and TANB77 clades are highlighted in red and orange, respectively. Color bars from the inner to the outer circles represent isolated status, phylum, and HiFi cMAGs, respectively. The numbers in parentheses represent the number of assembled cMAGs. **e** Number of cMAGs for each phylogenetic order. The text color represents phylum classification for each order. The plot uses the same color codes as in (c) except for the archaeal order Methanobecteriales (black).

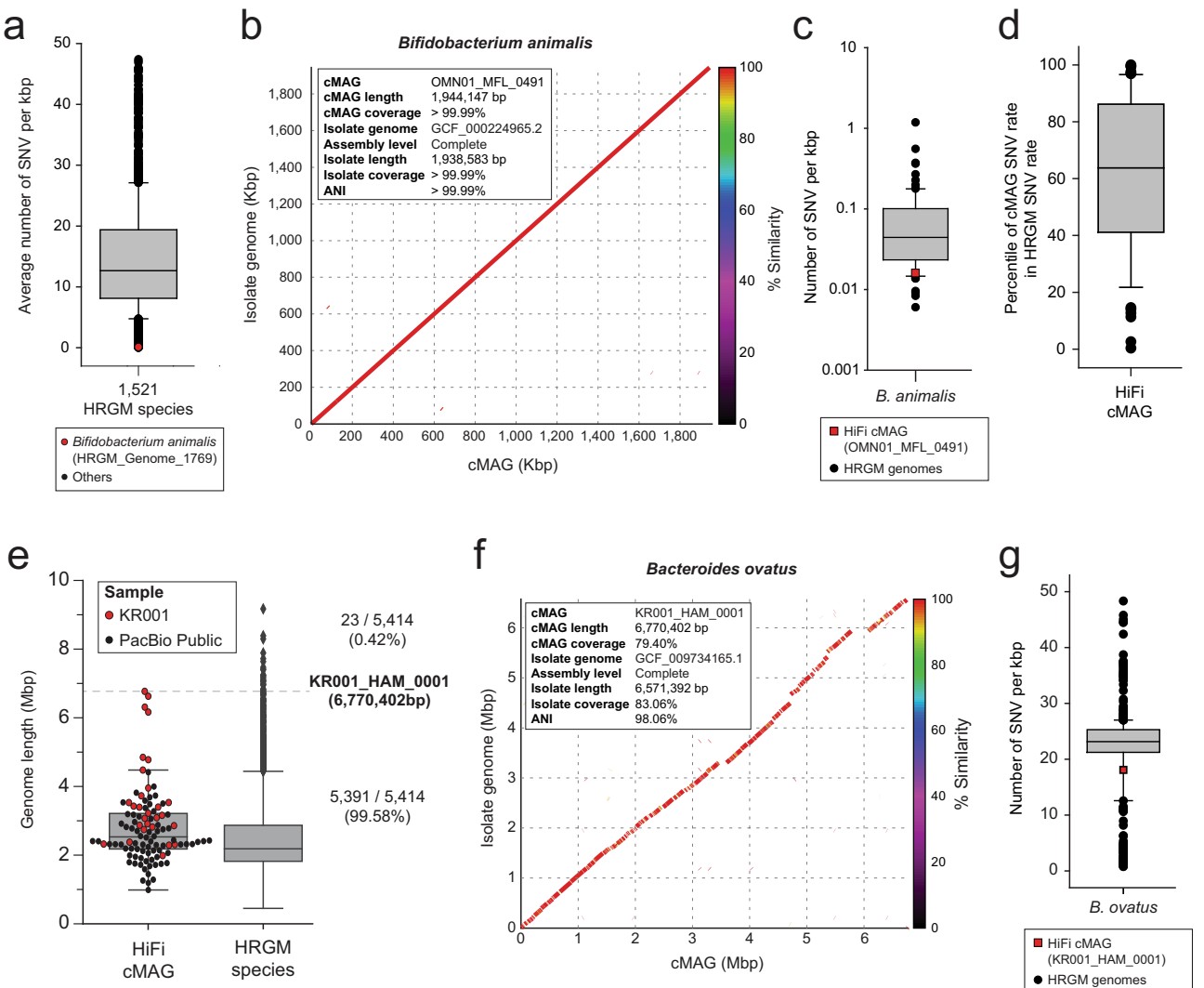

**Fig. 2 | HiFi metagenomic sequencing reconstructs accurate and long prokaryotic genomes. a** Single nucleotide variation (SNV) rate of HRGM species (*n* = 1521) and *Bifidobacterium animalis* (red dot). **b** Genome alignment plot between *B. animalis* cMAG (*x*-axis) and the closest isolated genome (*y*-axis). **c** SNV rate of *B. animalis* cMAG and its conspecific HRGM genomes (*n* = 102). **d** Percentile rank SNV rate of cMAGs (*n* = 77) among conspecific genomes. **e** Assembled genome length of cMAG (*n* = 102) and HRGM (*n* = 5414) species. The

horizontal dashed line represents the length of the longest HiFi cMAG. **f** Genome alignment plot between *Bacteroides ovatus* cMAG (*x*-axis) and the closest isolated genome (*y*-axis). **g** SNV rate of *B. ovatus* cMAG and conspecific HRGM genomes (*n* = 1056). The maximum and the minimum of the boxplots represent the 10th and 90th percentile of the data. The upper and lower bounds of the box represent the 25th and 75th percentile. The center bar represents the median. All the outliers are shown in the plots.

Supplementary Data 1). The longest cMAG (6,770,402 bp) was KR001_HAM_0001, which might correspond to *Bacteroides ovatus* and is larger than 99.58% of 5414 HRGM species in terms of genome size. The longest cMAG previously reported by ONT metagenomic sequencing was 3,825,229 bp[6]. Thus, KR001_HAM_0001 is the largest cMAG ever published. To confirm the integrity of KR001_HAM_0001, we compared it with the closest genome derived from isolate (RefSeq: GCF_009734165.1), to which it was highly similar in most genomic regions (ANI > 0.98 and alignment coverage ~0.80) (Fig. 2f). In addition, the SNV rate of KR001_HAM_0001 did not diverge significantly from that of its conspecific HRGM genomes (14% from the lowest) (Fig. 2g). These results suggest that HiFi metagenomic sequencing assembles accurate and complete genomes of human gut microbiota, including species with a genome size exceeding 6 Mbp.

### HiFi metagenomic sequencing assembles complete genomes for uncultured taxa

More than 80% of human gut microbial species are uncultured[2,3]. Anticipating the benefit of genome cataloging through cMAGs, we

surveyed previously entirely uncultured human microbial taxa. The culturability status of each cMAG was defined based on the presence of conspecific genomes derived from isolates in the HRGM, hGMB, and NCBI genome databases. This yielded 63 cultured cMAGs and 39 uncultured cMAGs (Fig. 3a, Supplementary Data 4). Among the 63 cultured cMAGs, 24 had discontinuous genomes with gaps and consisted of genomic scaffolds. Therefore, cMAGs obtained by HiFi metagenomic sequencing improved genome quality. Applying GTDB annotations, we identified cMAGs for uncultured taxa comprising 35 species, 19 genera, 4 families, and 2 orders (Fig. 3b, Supplementary Data 5).

Although the RF39 and TANB77 orders include 154 and 120 species, respectively (based on the HRGM), none of the species belonging to these taxa have been isolated so far. We identified three cMAGs belonging to RF39 and nine to TANB77. HiFi metagenomic sequencing assembled complete genomes for prokaryotic species of these entirely uncultured order-level taxa of the human gut microbiome. RF39 is a newly defined order of the Bacilli class, with more than 50% of species being classified neither at order level nor through consensus with NCBI

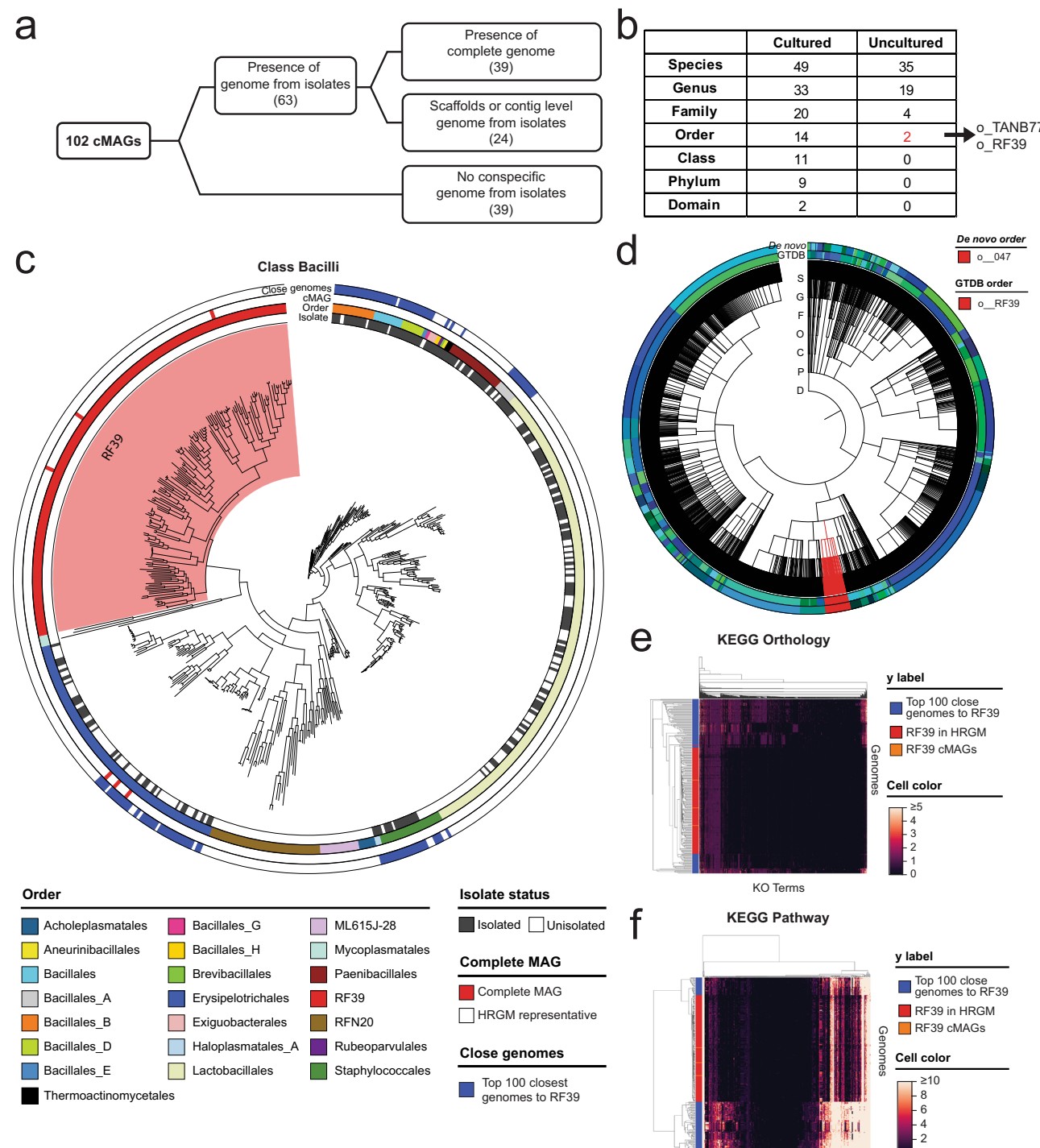

**Fig. 3 | Assembly of complete genomes for uncultured microorganisms.**
**a** Number of cMAGs with isolated and complete conspecific genomes. The 39 cMAGs for uncultured taxa are marked in red. **b** Number of cultured and uncultured taxa among the 102 cMAGs. **c** Maximum likelihood phylogenetic tree of HRGM species and HiFi cMAGs in the Bacilli class. The red-highlighted area represents the RF39 order. Isolated status, order, HiFi cMAG, and top 100 closest genomes to the RF39 order are annotated from the innermost to the outermost circle, respectively. **d** Phylogenetic tree of 4,545 bacterial species of HGM. The inner circle annotates the order according to the GTDB; whereas the outer circle represents the order according to the HGM *de-novo* classification. RF39 and o_047 orders are highlighted in red. **e, f** Hierarchically clustered heatmaps representing KEGG Orthology (**e**) and KEGG Pathway (**f**) profiles of 157 RF39 (red and orange rows) and the top 100 closest genomes to RF39 (blue rows).

taxonomic annotation (Supplementary Fig. 6a). Here, RF39 formed a distinct clade in the phylogenetic trees of Bacteria and Bacilli (Fig. 1d and Fig. 3c). Given that the phylogenetic tree and GTDB classification were generated based on the same bacterial marker proteins (bac120), we reviewed the de novo classification using an independent set of marker proteins[19]. Interestingly, all RF39 bacteria matched the de novo order o_047, and vice versa (Fig. 3d). Next, we predicted the microbial proteins of species belonging to the RF39 order and their neighboring species, and performed hierarchical clustering based on protein functional annotation. Most species of the RF39 order clustered

together, supporting their functional distinctiveness (Fig. 3e, f). TANB77 is another newly defined GTDB order, whose species are often classified into Clostridiales order by NCBI taxonomy (Supplementary Fig. 6b). Using the same procedure as for RF39, we confirmed the functional distinctiveness of the TANB77 order (Supplementary Fig. 6c–f). Together, these results imply that HiFi metagenomic sequencing enables the reconstruction of complete genomes belonging to uncultured prokaryotic clades of human gut microbiota.

### HiFi cMAGs reveal that hard-to-recover genomic regions are mostly genomic islands and rRNA

MAGs obtained by short-read metagenomic sequencing contain many gaps that represent hard-to-recover genomic regions[5]. Repetitive and mobile sequences negatively affect correct and continuous assembly. Given that cMAGs are circularized genomes with no gaps, alignment of conspecific genomes generated by short-read metagenomic sequencing allows the systematic search for hard-to-recover regions. Owing to the taxonomic diversity of the 102 cMAGs identified in this study, an unbiased taxonomic search was conducted. Quantification of retrieval rate of cMAG by conspecific HRGM genomes (Methods) revealed poorly recovered regions (Fig. 4a). With 20% intervals for bins of retrieval rate, most 1-kbp regions belonged to either >80% or ≤20% retrieval rates (Fig. 4b), indicating a clear separation between well and poorly covered regions.

Highly conserved sequences (e.g., rRNAs) and mobile sequences (e.g., genomic islands) are notoriously difficult to retrieve from short-read metagenomic assembly and to group into MAGs during the binning step[4,5]. Indeed, here, rRNAs and genomic islands aligned well with cMAG regions with low retrieval rate (lower than 20% for more than five consecutive 1-kbp genome bins) (Fig. 4a). In addition, 1-kbp genome bins containing rRNAs and genome islands exhibited significantly lower retrieval rate than other regions (Fig. 4c, d). Given that prokaryotic genomes have on average 4.2 copies of rRNAs[20] and the relatively short length of these sequences, genomic island regions seem to be the leading cause of gaps in MAGs obtained by short-read sequencing. We hypothesized that genomic island proteins could be more likely to be functionally characterized from cultured taxa with genomes derived from isolates. Indeed, genome island proteins of cultured taxa showed a higher eggNOG ortholog[21] annotation rate than uncultured taxa (Fig. 4e). Moreover, the read length of HiFi metagenomic sequencing samples was similar or larger than the median length of genomic islands (Fig. 4f). This suggests that HiFi reads likely cover genomic islands along with their adjacent regions, enabling correct mapping of foreign genetic elements to the host microbial genome.

## Discussion

HiFi sequencing offers substantial advantages in terms of base accuracy and read length[22]. In addition, HiFi metagenomic sequencing improves the quantity and quality of MAG assembly[10]. In the present study, we successfully retrieved complete circularized prokaryotic genomes of human gut microbiota through HiFi metagenomic sequencing without any binning process.

To take advantage of algorithmic complementarity, we utilized three different metagenomic assemblers: metaFlye, HiCanu, and hifiasm_meta. Initially, metaFlye generated three times more circular contigs than the other two assemblers, but most were filtered out, resulting in the lowest number of cMAGs. Instead, hifiasm_meta alone retrieved >88% of total cMAGs. Based on these results, we recommend hifiasm_meta for cataloging microbial genomes with HiFi metagenome sequencing.

Genome completeness is one of the most important criteria for cMAG. The completeness and contamination of MAGs are conventionally evaluated using single-copy genes[23]. This approach effectively filters out highly defective MAGs. However, single-copy gene-based completeness does not always coincide with actual completeness, especially for near-complete genomes. Therefore, we used the single-copy gene-based threshold to roughly filter out highly defective genomes, but not for the definitive evaluation of cMAGs. Moreover, lineage-specific marker proteins (e.g., checkM)[24] sometimes underestimate MAG completeness, particularly for novel clades, due to the incorrect use of clade-specific marker genes, explaining why we used universal single-copy genes bac120 or arc122. Indeed, using Clostridiales lineage-specific markers for checkM assessment, we attained 80% cMAG completeness for TANB77 genomes. For the same reason, the genomes of many uncultured taxa showed relatively low completeness by checkM. Notably, a recent update of the CheckM2 also addressed this issue[25]. Indeed, we observed increased completeness of TANB77 cMAGs with CheckM2 (Supplementary Fig. 7, Supplementary Data 7). The ideal way to assess MAG quality is through comparison with reference-quality conspecific genomes. However, this is not always possible, particularly for uncultured taxa. Therefore, we devised a novel method based on congruency with conspecific MAGs, which effectively filtered out circular contigs with gaps.

We assessed completeness of circular contigs using core contigs rather than core genes share among conspecific genomes. Conceptually, we may use core genes for the analysis of congruence of circular contigs with their conspecific genomes instead. However, gene-based methods need to conduct an additional process for gene calling from the circular contigs which requires high computational cost. Moreover, comparison with protein sequences of the core genes would be less intuitive than comparison with nucleotide sequences of core contigs. Thus, the contig-based congruence analysis provides computational advantages over the gene-based analysis.

No conspecific genomes derived from isolates were found for 39 HiFi cMAGs because they belonged to uncultured taxa. Notably, even 24 HiFi cMAGs belonging to cultured taxa possessed conspecific genomes derived from isolates consisting of discontinuous scaffolds. These results suggest that HiFi cMAGs can ameliorate existing human gut microbial genome catalogs not only for uncultured taxa but also for cultured ones. Many uncultured taxa have recently been defined through MAGs and have been newly classified by the GTDB[26]. These taxa often suffered from discordant annotation between the GTDB and NCBI classification systems. Indeed, the RF39 and TANB77 orders displayed distinct sets of marker proteins and functional profiles, supporting the GTDB as a reliable classification for uncultured taxa entirely composed of MAGs.

HiFi cMAGs enabled the unbiased examination of genomic regions poorly retrieved by short-read assembly. The latter contained highly conserved sequences (e.g., rRNA) and mobile sequences (e.g., genome islands), which likely caused fragmentation during MAG assembly. Highly conserved regions of 16S rRNA complicate assembly, especially of the short-read type. Even MAGs with high completeness often lack 16S rRNA regions. Genomic islands are important because they confer strain diversity as their sequences originate from other species. Given that many binning algorithms cluster contigs based on species-specific k-mer frequency, genome islands rarely cluster together with other contigs originating from the same genome. Importantly, HiFi cMAGs can circumvent this problem because they do not rely on binning during assembly.

There are limitations in our study. Our method for selecting cMAGs relies on the congruence with their conspecific genomes, but not all cMAGs have sufficient number of conspecific genomes for evaluation (e.g., ≥5 in this study). This requirement may penalize under-represented genomes and potentially result in the exclusion of novel species genomes. Indeed, we had to exclude two circular contigs that could not be evaluated for this reason. In addition, we tested only five HiFi metagenomic sequencing samples in this study. Therefore, some of the results (e.g., performance of each assembler) and the cMAGfilter parameters may not be generalized.

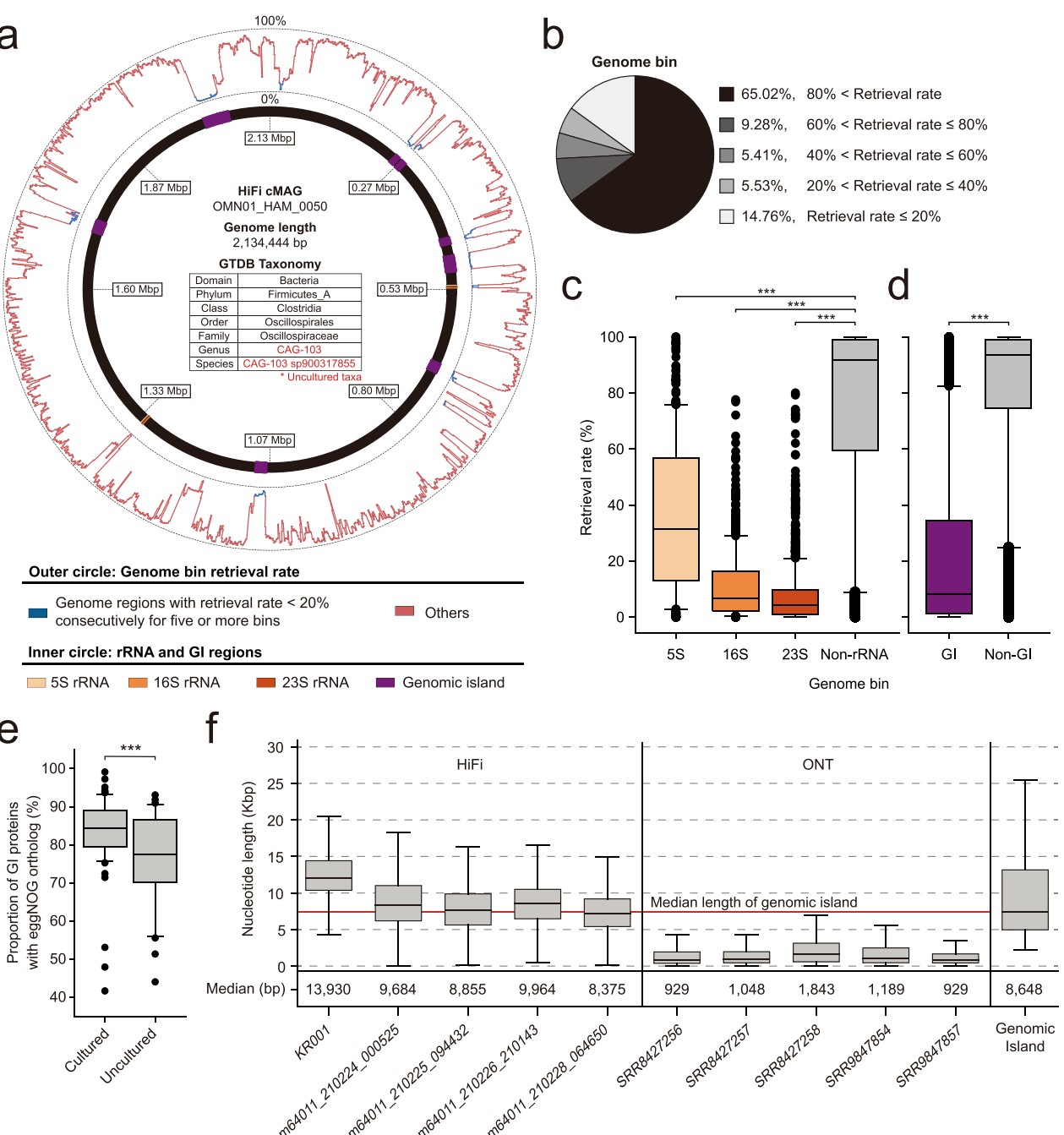

**Fig. 4 | HiFi-assembled cMAGs retrieve hard-to-assemble regions by short-read assembly. a** Genome bin retrieval rate and genome region annotation plot of a cMAG for the uncultured species OMN01_HAM_0050. The inner circle annotates rRNA and genomic island regions. The outer circle represents retrieval rate by conspecific HRGM MAGs for every 1-kbp genome bin. **b** Proportion of genome bins by retrieval rate. **c**, **d** Retrieval rate of genome bins that include rRNAs (**c**) or genome islands (**d**) (n = 393 for 5S rRNA genome bins, n = 662 for16S rRNA genome bins, n = 1258 for 23S rRNA genome bins n = 283,267 for non-rRNA genome bins, n = 30,330 for GI genome bins, and n = 255,146 for non-GI genome bins). Retrieval rate was compared by the two-sided Mann–Whitney U test. (P-value = 1.83e−97 for 5S rRNA, P-value = 6.39e−286 for 16 S rRNA, P-value < 1e−300 for 23S rRNA compared to non-rRNA genome bins; P-value < 1e−300 for GI genome bins compared to non-GI genome bins). **e** Proportion of genome island proteins with an eggNOG ortholog according to the cultured status

of the host genome (n = 63 cultured genomes, n = 38 uncultured genomes). The proportion was compared by the two-sided Mann–Whitney U test (P-value = 0.0009). **f** Read length of long-read human fecal metagenomic samples and length of entire genome islands found among 102 cMAGs. (n = 2,017,709 reads for KR001, n = 1,792,146 reads for m64011_210224_000525, n = 1,687,238 reads for m64011_210225_094432, n = 1,904,159 reads for m64011_210226_210143, n = 1,767,289 reads for m64011_210228_064650, n = 4,487,361 reads for SRR8427256, n = 3,404,101 reads for SRR8427257, n = 4,003,722 reads for SRR8427258, n = 11,016,028 reads for SRR9847854, n = 10,261,578 reads for SRR9847857). The maximum and the minimum of the boxplots represent the 10th and 90th percentile of the data. The upper and lower bounds of the box represent the 25th and 75th percentile. The center bar represents the median. All outliers are shown in (**c**–**e**) and omitted in (**f**). (***P-value < 0.001).

While this manuscript is being peer reviewed, a public release of nine cMAGs assembled by HiFi sequencing was announced by independent research group[27]. However, the study did not present their systematic evaluations on nucleotide accuracy, taxonomic diversity, and ability in large genome assembly, probably due to the insufficient number of retrieved cMAGs.

In summary, we expect that HiFi metagenomic sequencing will facilitate cataloging accurate complete genomes of microbiota. Extensive application of HiFi metagenomic sequencing of human fecal samples would improve our understanding of the human gut microbiome. Although HiFi metagenomic sequencing is relatively expensive and requires high molecular weight DNA, these limitations will be likely overcome by future technical improvements.

## Methods

### HiFi metagenomic sequencing data from human fecal samples

We collected four public HiFi metagenomic sequencing samples of human fecal samples recently released by Pacific Bioscience. The samples were collected from US and pooled before the sequencing. Two of them were from pooled fecal samples of vegan donors, and the others were from pooled fecal samples of omnivore donors. In addition, we generated in-house HiFi metagenomic sequencing data from a fecal sample provided by a Korean donor (Male, 55 years old) following full informed consent and approval by the Yonsei University Institutional Review Board (IRB No. 4-2020-0309). The donor's entire metagenomic DNA was sequenced by Macrogen Inc. (Republic of Korea). A high-quality and high-molecular weight genomic DNA sample, in which most fragments exceeded 40 kbp, was determined by pulsed-field gel or capillary electrophoresis and used for HiFi SMRTbell library preparation. The concentration of genomic DNA was measured by Pico-Green and its quality was evaluated in a Femto Pulse system (Agilent). We used 8 μg of input genomic DNA for HiFi library preparation. Femto Pulse helped determine the size distribution of genomic DNA < 15 kbp. Genomic DNA > 40 kbp was sheared by Megarupor3 and purified using AMPurePb magnetic beads. A HiFi SMRTbell library (10 μL) was prepared with the PacBio SMRTbell Express Template Prep Kit 2.0, annealed using the Sequel II Bind Kit 2.2 and Internal Control Kit 1.0.0, and sequenced with the Sequel II Sequencing Kit 2.0 and Sequel II SMRT cell 8 M Tray. A 30-h video was recorded in each SMRT cell using the Sequel II platform. Subsequent procedures were performed according to the PacBio SampleNet-Shared protocol.

### Removal of host contaminants and de novo genome assembly of HiFi sequencing reads

We aligned all HiFi reads against the human reference genome (GRCh38.p13) with minimap2 v2.18-r1015[28] aligner using the -ax asm20 parameter. Reads aligned to the human reference genome were considered human contaminants and were disregarded in downstream analysis. HiFi sequencing reads were assembled using three different metagenomic assemblers: HiCanu v2.1.1[13], metaFlye v2.8.3-b1695[12], and hifiasm_meta v0.2-r040[11]. For HiCanu, we used the -pacbio-hifi mode with recommended maxInputCoverage = 10000, corOutCoverage = 10000, corMhapSensitivity = high, corMinCoverage = 0, and genomeSize = 3.7 M. For metaFlye, we applied the –pacbio-hifi –meta options. We ran hifiasm_meta with default parameters and used primary contigs.

### Filtering circular contigs based on biological priors

Only circular contigs obtained by metagenome-assembly entered our bioinformatics workflow to select cMAGs. We first filtered out ambiguous or non-prokaryotic circular contigs based on the assembly structure and prior biological knowledge. Contigs with assembly bubbles or repeats were excluded. The HiCanu assembler provides assembly bubbles and repeat annotations. For the contigs by metaFlye and hifiasm_meta assemblers, we identified repeats and bubbles from

the output Graphical Fragment Assembly file using BubbleGun v1.1.1 software. Contigs shorter than 100 kbp were also removed, as they were highly unlikely to correspond to a complete prokaryotic genome (Supplementary Fig. 2a). The remaining contigs were evaluated for completeness based on ubiquitous single-copy marker proteins. We predicted 120 bacterial (bac120) and 122 archaeal (arc122) marker proteins using GTDB-Tk v1.6.0[29] and filtered out circular contigs with fewer than 100 of these marker proteins. Because marker counts gradually decreased before the threshold and dropped swiftly thereafter, faulty circular contigs could be distinguished (Supplementary Fig. 2b). We also examined the existence of rRNAs (5S, 23S, and 16S rRNA) and tRNAs. We predicted rRNAs with barrnap v0.9 tools using the –kingdom bac and –kingdom arc parameters according to taxonomy by GTDB-Tk. We applied an evalue of 1e−04 for 5S rRNA (given its shorter length) and an evalue of 1e−06 (default) for the 16S and 23S rRNAs. We excluded contigs lacking rRNAs. We predicted tRNAs using tRNAscan-SE v2.0.7[30] with B and A parameters for bacterial and archaeal contigs, respectively. We considered non-pseudo tRNAs only and disregarded circular contigs with fewer than 20 tRNA types. Circular contigs that passed each filtering step are listed in Supplementary Data 3.

### Removal of redundant circular contigs

The filtered circular contigs could be redundant because they were assembled from pooled samples. Therefore, we used sequence similarity to remove redundant circular contigs. We aligned every pair of circular contigs with nucmer 4.0.0beta2[31] and used a delta-filter program to identify the best-bidirectional alignments. When we plotted the contig pairs by ANI (identical sequence length/aligned sequence length) and maximum alignment coverage (aligned sequence length/ shorter contig length), we found a cluster of highly similar contigs with ANI > 0.99 and maximum alignment coverage > 0.95 (Supplementary Fig. 2d). Most pairs within this group consisted of contigs from the same or the same diet type of sample (Supplementary Fig. 2e). In addition, when we sorted the contig pairs by ANI multiplied by the maximum alignment coverage (similarity index), there was a clear discrimination between contig pairs above or below the similarity index of 0.9405 (Supplementary Fig. 2f). Therefore, we determined ANI > 0.99 and maximum alignment coverage > 0.95 as the thresholds that removed redundant contigs while maintaining strain diversity. We selected a contig with the largest sum of ANI as a representative for each cluster of redundant contigs (Supplementary Fig. 2g, Supplementary Data 2).

### Filtering circular contigs based on congruency with conspecific genomes

Filtered circular contigs by biological priors may still contain genomes with gaps. To identify such faulty genomes, we conducted the following processes using the HRGM catalog[3]: (i) finding conspecific HRGM genomes, (ii) identifying core contigs shared by most conspecific genomes, and (iii) aligning core contigs to the query circular contig (Supplementary Fig. 3a). For each non-redundant circular contig, we found its conspecific genomes from the HRGM non-redundant genome set. We adopted a reduced search strategy because aligning all circular contigs against every non-redundant genome requires excessive computer power. We first aligned each circular contig against HRGM species representatives using nucmer with the –mum option followed by a delta-filter with -r and -q options. Only HRGM species representatives within the same genus as the circular contig by the GTDB taxonomy annotation were considered because genomes of other genera were highly unlikely to meet the conspecific identity threshold (ANI > 0.95)[2,3,19,32,33]. Based on the species with the highest similarity index (ANI × maximum alignment coverage), we performed genome alignment using nucmer for all non-redundant HRGM genomes. By definition, HRGM conspecific genomes of the circular contig

had ANI > 0.95 and maximum alignment coverage > 0.8. For three circular contigs (VEG02_HAM_0051, OMN01_HAM_0037, and OMN01_HAM_0001) within the *Collinsella* genus, which are reported to have an exceptionally high variant rate[3], we adjusted the ANI threshold to 0.94 to find sufficient conspecific HRGM genomes for further analysis. We stopped searching when we found 100 conspecific genomes for each circular contig, while dismissing two circular contigs with fewer than five conspecific HRGM genomes.

Most conspecific genomes are MAGs, which are fragmented and may contain contamination. Here, we hypothesized that contigs shared by most conspecific MAGs (core contigs) were likely complete genome sequences. Therefore, to find core contigs, we performed all pairwise genome alignments for the conspecific genomes of each circular contig using nucmer and delta-filter (same as above). For every contig of a conspecific genome, we considered that the contig was present in the other conspecific genome when more than half of the contig sequences were aligned with >95% identity. A core contig was defined as longer than 5 kbp (to exclude short sequences which tend to be more vulnerable) and present in more than 80% of conspecific genomes (to select highly shared contigs).

Finally, we aligned the core contigs to the query circular contig. Because the core contig set could contain redundant sequences, we used nucmer with the *–maxmatch* option. We calculated the core contig retrieval rate (aligned core contigs/core contigs) for each circular contig and excluded six circular contigs with core contig retrieval rate <0.95 (Supplementary Fig. 3b). A total of eight circular contigs (two with small conspecific genome count and six with low core contig retrieval rate) were excluded from the final list of cMAGs.

### Evaluation of cMAGs by the GC-skew pattern
For the final 102 cMAGs, we calculated and plotted the GC-skew pattern of each contig using the *gc_skew.py* script in the iRep[34] package (1 kbp window, 10 bp slide). Next, we manually divided the cumulative GC-skew patterns into five classes: (i) very clear (17 cMAGs), whereby the curve showed apparent symmetry with respect to the Ter site and was almost linear; (ii) clear (14 cMAGs), whereby the curve displayed clear symmetry but also a few minor non-Ter site peaks (few thousand bp); (iii) decent (53 cMAGs), whereby the most significant peak of the curve was the Ter site, but several minor peaks also existed; (iv) poor (10 cMAGs), whereby the curve exhibited distinct ascending and descending regions but no symmetry or had significant peaks other than the Ter site; and (v) no-pattern (8 cMAGs), whereby the curve presented no GC-skew pattern for Ter or Ori sites. The GC-skew class annotation is provided in Supplementary Data 4, and GC-skew plots are presented in Supplementary Fig. 4.

### SNV analysis
We aligned cMAGs to HRGM species representatives and found conspecific HRGM species with ANI > 0.95 and maximum alignment coverage > 0.6. For cMAGs with multiple conspecific HRGM species, we selected only the species with the highest similarity index (ANI × maximum alignment coverage). To obtain a reliable SNV density percentile value of cMAG, we performed subsequent SNV analysis only if there were >100 conspecific genomes for the HRGM species cluster. We aligned cMAG and HRGM genomes (non-representative) against representative HRGM species using nucmer, and filtered the best bidirectional alignments using a delta-filter with -r and -q options. Next, to eliminate indels, SNVs were identified using *show-snps* in the *mummer* package with -I options. The overall procedure is depicted in Supplementary Fig. 5.

### Conspecific isolates-derived or complete genomes for cMAGs
We compiled genomes derived from isolates sequences from HRGM[3], hGMB[35], and the recently updated NCBI genome databases[17,36]. Then, we aligned cMAGs to genomes derived from isolates with nucmer and

delta-filter, as described in the redundancy removal step. We defined cMAGs with conspecific genomes derived from isolates as having at least one genome derived from isolates, ANI > 0.95 and maximum alignment coverage > 0.6. For such cMAGs, we manually searched whether the conspecific isolate was complete in the NCBI genome database.

### Reconstruction of the phylogenetic tree
We predicted bac120 or arc122 proteins using the GTDB-Tk identify module[29] and performed multiple sequence alignment with the GTDB-Tk align module. The tree was calculated using IQ-Tree v2.1.3[37] and visualized with ITOL v6[38].

### Examining the taxonomic distinctiveness of RF39 and TANB77 orders
To investigate the phylogenetic distinctiveness of RF39 and TANB77 orders, we gathered HRGM species and cMAGs within the Bacilli and Clostridia classes, which were their respective parental clades. We then constructed a phylogenetic tree of Bacilli and Clostridia, and selected the top 100 closest genomes of RF39 and TANB77 orders based on the average maximum likelihood distance. For RF39 and TANB77 orders and each of their close neighboring genomes, we predicted their proteins with Prokka[39] and annotated protein functions with eggNOG-mapper v2.1.6[40]. Hierarchical clustering was performed using KEGG Orthology and KEGG Pathway[41] profiles based on Euclidean distance.

### Genome bin retrieval rate by the short-read assembled genome catalog
We measured the retrieval rate of cMAGs to identify hard-to-recover genome regions by short-read assembly. First, we aligned conspecific genomes to each cMAG using nucmer, and identified the best alignments with delta-filter using the -r parameter. We then calculated the genome bin retrieval rate for every 1-kbp bin, $B$, of cMAG as follows:

$$Retrieval\,rate_B = \frac{\sum\limits_{p \in B} matchedcount_p}{1,000 \times \#\,of\,conspecific\,genomes} \times 100(\%)$$

where $p$ indicates the single nucleotide position in $B$.

To investigate the characteristics of low retrieval rate regions, we compared the retrieval rate of genome bins containing rRNAs or genomic islands with other regions. We predicted the 5S, 16S, and 23S rRNA regions using barrnap (as described in the first filtration section). In addition, we annotated cMAGs with Prokka v1.14.6[39] and identified genome island regions using IslandViewer 4[42]. Proteins located within genome islands were predicted by prodigal v2.6.3[43] and we annotated their eggNOG ortholog and function using eggNOG-mapper 2[40].

### Reporting summary
Further information on research design is available in the Nature Research Reporting Summary linked to this article.

## Data availability
The public HiFi metagenomic sequencing data for pooled human fecal samples are available from NCBI Sequence Read Archive under the accession code PRJNA750084. HiFi metagenomic sequencing data for a Korean fecal sample generated in this study have been deposited in the Sequence Read Archive under the accession code PRJNA798244. The entire 102 cMAGs generated in this study have been deposited in the GenBank under the accession code PRJNA798244. The GC-skew and genome bin retrieval rate plots for entire 102 cMAGs are available at https://doi.org/10.5281/zenodo.5996768. The short-read sequencing based human gut microbiome genome catalog is available from the HRGM database (https://www.mbiomenet.org/HRGM/). The additional list of cultured genomes is obtained from the hGMB database

(https://hgmb.nmdc.cn/). Human reference genome is downloaded from NCBI-Assembly (https://www.ncbi.nlm.nih.gov/assembly/GCF_000001405.39/).

## Code availability

Source code of cMAGfilter that filters out defective circular contigs based on congruency with conspecific genomes is available at https://github.com/netbiolab/cMAGfilter.

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

## Acknowledgements

This research was supported by the National Research Foundation (NRF) & funded by the Korean government (MSIT) (No. 2018R1A5A2025079, 2019M3A9B6065192, and 2022M3A9F3016364 to I.L.) and supported in part by the Brain Korea 21 (BK21) FOUR Program to I.L. HiFi sequencing was provided by the SMRT Grant of MdxK, Macrogen, and PacBio to I.L.

## Author contributions

C.Y.K. and I.L. conceived and designed the study. C.Y.K. and J.M. performed the bioinformatics analysis on metagenomic data and formulated the study hypothesis. I.L. supervised the bioinformatics analysis. All authors contributed to the writing of the manuscript.

## Competing interests

The authors declare no competing interests.

## Additional information

**Correspondence and requests** for materials should be addressed to Insuk Lee.

