## [Peer Review File · Nature Communications]

REVIEWER COMMENTS

Reviewer #1 (Remarks to the Author):

The manuscript by Kim and colleagues presents a novel approach exploiting long-reads technologies to retrieve almost complete metagenome-assembled genomes.

Although this is of interest as recent works in the microbiome field are heavily exploiting MAGs to uncover and study also the yet-to-be characterized fraction of the microbiome, several aspects should be better clarified and verified to ensure these MAGs are actually of high quality.

For instance:

- Line 27, "Nucleotide accuracy of the final cMAGs was similar to that of Illumina", here similar is reflecting that this approach is not better than more common short read-based approaches. Could this be due to possible higher error rates of long reads and/or less coverage due to less amount of throughput?

- Line 28, "The cMAGs could exceed 6 Mbp", is this something expected and/or desirable? It feels like long genomes might indicate a higher contamination rate.

- Lines 51-52, "Seeking only circularized contigs, we skipped the binning procedure.", it is not clear the reason why not applying also the binning step. Since only 5 samples were analyzed, it could be worth looking also, other than circular contigs, you can retrieve more assemblies that could be almost complete, even though not circular exploiting long-read sequencing,

- Line 53, "for authentic complete prokaryotic genomes.", how do you assess authenticity and completeness? Especially for the two orders that are uncultured?

- Lines 54-56, "microbiota. We demonstrate that cMAGs obtained by HiFi metagenomic sequencing share similar nucleotide accuracy as Illumina short-read-based MAGs.", so, HiFi assemblies retrieve the same MAGs we would have out of short-reads assemblies? Is there anything that long-reads can assemble and short-reads can't?

- Lines 56-67, as per the previous comment, is this 6Mbp something expected/desirable?

- Lines 105-106, "Overall, our bioinformatics workflow retained 102 of the 3,354 circular contigs initially assembled.", it will be fairer to compare the 102 with the total number of circular contigs filtered from the previous step (145+76+52 - those found identical).

- Lines 107-116, although that could be the case, instead of GC often the tetra-nucleotide frequencies are compared, have you checked them?

Others:

- Line 23, "Recently, nanopore metagenomic sequencing assembled 20 complete MAGs", it's not nanopore that assembled MAGs, please consider rephrasing for clarification.

- Line 25, "high-accuracy long-read (HiFi)", it is not very clear what HiFi is an acronym for

-

Reviewer #2 (Remarks to the Author):

In this study, the authors Kim et al. report a new list of complete MAGS (cMAGs) assembled from five human fecal samples, sequenced with PacBio HiFi technology. While the samples and methodologies used are not novel, the workflow provided by the authors to detect and filter out biased cMAGS is interesting, combining multiple quality control methodologies from marker genes detection to GC bias exploration.

As the actual state of the paper is providing an interesting workflow but missing clarity in the pipeline and closely related work that used the same samples, I think that the paper should be accepted after incorporating the following suggestions:

Major comments:

1. Most samples used in this study were already used to build MAGs by Feng et al (<https://doi.org/10.1038/s41592-022-01478-3>). Consequently, this study provides a resource with limited novelty. Comparison with MAGs published by Feng et al. would be welcomed.
2. The workflow created by the authors is interesting, but raises multiple questions. For example, if I understand correctly, one step of the filtering remove all contigs with less than 5 conspecific HRGM genomes and with less than 95% core contig retrieval. These parameters seem penalizing for under-represented genomes, or genomes that are highly distributed in the human population that include geographic microbial disparities. This will for instance favor species with lower diversity, such as the cMAG used as validation GCF_000224965, which is in fact a probiotic bacterium widely used in food industry (see DOI: 10.1128/JB.06079-11). The authors should then discuss the necessity to use core contigs, ie contigs >5kbp long that are similar in 80% of conspecific genomes, instead of the core genes only. Also, are the samples pooled prior to assembly ? It would be nice to make this more transparent in the Fig 1.
3. 7 complete MAGs were recovered with HiCanu only. Contrary to the two others assemblers, HiCanu was not designed to process metagenomic data. Could the authors explain why these CMAGs were missed by Hifiasm-meta and metaflye?
4. "Genomic island proteins are more likely to be discovered from cultured taxa with isolated genomes." "Thus, we hypothesized that cMAGs would provide opportunities to discover novel genome island genes." These statements are probably wrong. If a region is missing in a MAG, it does not mean it could not be assembled. Contigs corresponding genomic island are present in metagenomic assemblies obtained with short-reads. Yet, they are missing in MAGs because binning tools often miss them.

5. Similar studies previously published are not cited but cover the same topic:

<https://doi.org/10.1080/19490976.2021.2021790>

<https://doi.org/10.1128/mra.00250-22>

<https://doi.org/10.1038/s41592-022-01478-3>

Minor comments:

1. l. 21 : replace “discontinuity” by “fragmentation”
2. l. 26 : replace “were filtered » by “were selected”
3. l. 26,79 : “authentic prokaryotic genomes” this term is too vague. Please rephrase.
4. l. 29,30 : “whose genomes have not been characterized yet.” This is not true:
<https://doi.org/10.1186/s12864-020-06807-4>
5. l. 32 : “of human gut microbiota” This may be true for other microbial communities
6. l.35: replace “culturomics technology” by “culturomics techniques”
7. l. 41: genome fragmentation is a consequence of “conserved, repetitive, and mobile sequences”, but not chimerism
8. l. 49 : HiCanu was not designed for metagenomic assembly. Please cite MetaFlye. The name of the assemblers should be explicitly written
9. The end of the introduction is the abstract the paper.
10. l.67 & l.277 The URL does not work. The authors should provide a resource available on the INSDC
11. L119 : “Although bias toward a few bacterial clades (e.g., Oscillospirales) was observed” Is this bias related to the estimated taxonomic composition of samples?
12. L126: “same-strains from different samples are hardly identical to each other because of genetic variation accrued via horizontal gene transfer and spontaneous mutation.” I don’t understand this sentence. Same strains are supposed to be identical. Please rephrase.
13. L129: The chosen CMAG probably derives from a probiotic strain which explains the very high similarity with GCF_000224965.2. I would mention it explicitly.
14. L. 174 This is not true. Plaza Onate et al. (doi: 10.1128/mra.00250-22) published complete MAGs for these two orders.
15. L. 193: Please, provide a reference (<https://doi.org/10.1099/mgen.0.000436> for instance)
16. L199 The term “genome coverage” is ambiguous and might be confounded with sequencing coverage. Please define the terms more precisely.
17. L.215 “ONT metagenomic sequencing samples did not reach the median genome island length” ONT can achieve longer read length by changing some parameters when building sequencing libraries. I don’t think this statement is relevant
18. L228 : metaFlye should perform well with HiFi metagenomic sequencing data. However, it might not handle pooled samples that contain multiple closely related strains as well as hifi asm meta.
19. L242 This issue is addressed in checkm v2. The authors could mention it.

20. L271 It is not a matter of abundance (i.e concentration) but a matter of quantity (i.e mass).

21. l.276: replace “datasets” by “samples”

22. Fig1B Replace the Venn Diagram with an upsetR figure

Responses to referees (our responses in bold)

NCOMMS-22-24815

HiFi Metagenomic Sequencing Enables Assembly of Accurate and Complete Genomes from Human Gut Microbiota

We would like to thank the editor and all referees for your helpful comments on this manuscript. We deeply appreciated the input, and we have enclosed revised version of the manuscript, Fig. 1, Fig. 4, Supplementary Table S7 (previously S6), and newly generated Supplementary Fig. 7 and Supplementary Table S6. We hope those will address the concerns in the reviews, to which we detail our response below.

=====

REVIEWER COMMENTS

Reviewer #1 (Remarks to the Author):

The manuscript by Kim and colleagues presents a novel approach exploiting long-reads technologies to retrieve almost complete metagenome-assembled genomes.

Although this is of interest as recent works in the microbiome field are heavily exploiting MAGs to uncover and study also the yet-to-be characterized fraction of the microbiome, several aspects should be better clarified and verified to ensure these MAGs are actually of high quality.

For instance:

- Line 27, "Nucleotide accuracy of the final cMAGs was similar to that of Illumina", here similar is reflecting that this approach is not better than more common short read-based approaches. Could this be due to possible higher error rates of long reads and/or less coverage due to less amount of throughput?

In fact, as demonstrated in Fig. 2a-d, the nucleotide accuracy of the final cMAGs by HiFi long read sequencing was as high as that of Illumina short read sequencing. Then, the word "similar" might confuse the readers. So we have modified the text as follows.

"Nucleotide accuracy of the final cMAGs was as high as that of Illumina"

Evaluation of the effect of the throughput on sequencing error is challenging especially with the short-read-based MAGs because MAGs were assembled from different samples with different sequencing depths. Also, the abundance of the species would significantly affect the sequencing coverage of the MAG.

We assessed the nucleotide accuracy of cMAGs by HiFi long read sequencing by comparing with their conspecific genomes assembled by short read sequencing, and thought the following aspects could also influence the result.

- 1. When there are multiple conspecific short-read MAGs from the same individual (e.g. longitudinal studies), the accuracy of the short read-based MAGs could be overestimated.**

2. When the conspecific genomes are selected from the isolated sample, which is relatively easier to assemble a more accurate genome, the accuracy could be favorably biased toward the short-read sequencing technique.

Considering all the aspects above, we believe that our assessment based on the *Bifidobacterium animalis* genome would be a useful resource to understand the nucleotide accuracy of HiFi cMAG (manuscript line 126-142). The species has the lowest SNV rate among all cultured species; therefore, we were able to control the effect of spontaneous mutation as much as possible. Compared to the RefSeq isolated genome of *B. animalis* (GCF_000224965.2), the nucleotide sequence of HiFi cMAG for the same species (OMN01_MFL_0491) was almost identical (ANI > 99.99% and alignment coverage >99.99%).

- Line 28, "The cMAGs could exceed 6 Mbp", is this something expected and/or desirable? It feels like long genomes might indicate a higher contamination rate.

It is only recently that the HiFi long-read sequencing is applied to complex metagenomic samples. Therefore, the number of available samples is limited and there are not many studies that could be used to estimate the maximum length of circularized MAGs with HiFi metagenomic sequencing technique. Therefore, we would not say it was an expected result.

However, we found that the longest cMAG from this study (KR001_HAM_0001) was longer than the 99.58% of the genomes of HRGM, a comprehensive human gut microbial genome catalog (Fig. 2e). This result suggests that the HiFi sequencing technology may be able to assemble circularized complete MAGs for most of the gut bacterial species (strictly, in terms of genome size). Therefore, we would say it is a desirable result.

We understand the reviewer's concern that the longer assembly could have the more contamination. However, we could not observe correlation between the cMAG length and the contamination (see the figure below). This also suggest that our computational workflow successfully filters out the long cMAGs with high contamination rate.

- Lines 51-52, "Seeking only circularized contigs, we skipped the binning procedure.", it is not clear the reason why not applying also the binning step. Since only 5 samples were analyzed, it could be worth looking also, other than circular contigs, you can retrieve more assemblies that could be almost complete, even though not circular exploiting long-read sequencing,

We agree that we could assemble more high-quality MAGs (completeness > 95% and contamination < 5%) by incorporating the binning process. However, in the present study, we focus on the investigation of complete and circularized MAG, which is a single circular contig rather than a bin of multiple contigs. We committed to develop the methodology to select complete prokaryotic genomes out of the single molecular circular contigs and investigate their properties such as nucleotide accuracy, taxonomy diversity and ability in large genome assembly. Therefore, we deliberately excluded the binning process which does not fit our purpose of the study.

Although not reporting in the manuscript, we performed the binning process with these HiFi assembly data during the study and acquired 73% more MAGs with high-quality per sample on average (see the figure below). We have a plan to include the additional high quality MAGs in our next version of the human gut microbiome catalog (HRGM v2).

- Line 53, "for authentic complete prokaryotic genomes.", how do you assess authenticity and completeness? Especially for the two orders that are uncultured?

First of all, we apologize for the confusing word ‘authentic’ in the text. We originally used the word to stress that we excluded any circular contigs other than complete prokaryotic genomes (e.g., plasmids, viral genomes, and etc.). But we realized that ‘complete prokaryotic genomes’ already deliver the sufficient meaning and extra use of the word ‘authentic’ only causes confusions. Thus, we decided to remove the word ‘authentic’ in the revised manuscript.

We assess completeness of circular contigs by leveraging congruency with their conspecific genomes using a software, cMAGfilter. This approach is robust and unbiased to the genome's cultured status. We searched at most one hundred conspecific genomes from the HRGM catalog and identified the core contigs shared

among >80% of the conspecific genomes. These highly-shared contigs are likely to be the real DNA fragments of the given species. And then, we filtered out circular contigs that miss more than 5% of the core contigs. We also demonstrated that the workflow successfully filters out the defective genomes using the taxa with reference-quality cultured genomes (Supplementary Fig. 3c-f).

- Lines 54-56, "microbiota. We demonstrate that cMAGs obtained by HiFi metagenomic sequencing share similar nucleotide accuracy as Illumina short-read-based MAGs.", so, HiFi assemblies retrieve the same MAGs we would have out of short-reads assemblies?

The sentence means that the nucleotide accuracy of the cMAGs (evaluated by SNV rate) by HiFi sequencing was as high as that of the short-read assembled MAGs. We rephrased this sentence for better clarity.

"We demonstrated that the nucleotide accuracy of HiFi cMAGs was as high as that of the short-read-based MAGs."

- Is there anything that long-reads can assemble and short-reads can't?

We appreciate the reviewer's insightful question. In fact, we could find one circular contig (VEG02_HAM_0059) that pass the first filtration step of our computational workflow but has no conspecific genomes in the HRGM (i.e. the genome of this species assembled by short-read sequencing has not been reported yet). This circular contig could represent either a novel species genome or a simply defective genome. Finally we decided to exclude this circular contig from the final list of cMAGs, because of the following reasons.

- 1. Given that the number of metagenomic samples used for cataloging HRGM genomes (over 12,000 samples) is much larger than the number of samples used in this study (5 samples), the probability of finding a novel species genome in this study is quite low. Therefore, the cMAG is more likely to be a defective genome.**
- 2. We could not assess the completeness of the circular contig, because our computational method, cMACfilter, requires conspecific genomes.**

We agree that our method for cMAG assessment is robust but could miss some novel genomes. Therefore, we have added a paragraph describing the limitation of our method in the discussion section.

"Our method for selecting cMAGs relies on the congruence with their conspecific genomes, but not all cMAGs have sufficient number of conspecific genomes for evaluation (e.g., ≥ 5 in this study). This requirement may penalize under-represented genomes and potentially result in the exclusion of novel species genomes. Indeed, we had to exclude two circular contigs that could not be evaluated for this reason."

- Lines 56-67, as per the previous comment, is this 6Mbp something expected/desirable?

As we answered your similar question for Line 28, it was not an expected but a desirable result.

- Lines 105-106, "Overall, our bioinformatics workflow retained 102 of the 3,354 circular contigs initially assembled.", it will be fairer to compare the 102 with the total number of circular contigs filtered from the previous step (145+76+52 - those found identical).

In the sentence, the "Overall, our bioinformatics workflows" does not only indicate the cMAGfilter, but also refers the entire process including the first step of filtration based on the biological prior and redundancy removal (see Fig. 1a). Given our intention for the sentence, the current text might be correct.

- Lines 107-116, although that could be the case, instead of GC often the tetra-nucleotide frequencies are compared, have you checked them?

We guess there was a confusion between GC-skew test for assembled genome quality and GC content analysis for binning which often uses tetra-nucleotide frequencies instead. Given that many prokaryotic genomes show symmetry of cumulative GC-skew with the axis of origin of replication, the GC-skew graph is suggested to assess the assembly quality of prokaryotic genomes (doi:10.1101/gr.258640.119). We checked the GC-skew to assess cMAG quality. To the best of our knowledge, there is no method for assessing genome assembly quality based on tetra-nucleotide frequencies.

Others:

- Line 23, "Recently, nanopore metagenomic sequencing assembled 20 complete MAGs", it's not nanopore that assembled MAGs, please consider rephrasing for clarification.

We rephrased the sentence as suggested.

"Recently, 20 complete MAGs (cMAGs) have been assembled from Oxford Nanopore long-read sequencing of 13 human fecal samples,"

- Line 25, "high-accuracy long-read (HiFi)", it is not very clear what HiFi is an acronym for

Originally, HiFi is an abbreviation of 'Hi'gh-'Fi'delity. However, the technology provider, PacBio, commonly refer to as "highly accurate" long-read sequencing. We also found other publications that describe the HiFi technology as highly accurate long-read sequencing. (<https://www.nature.com/articles/s41597-020-00743-4>)

Reviewer #2 (Remarks to the Author):

In this study, the authors Kim et al. report a new list of complete MAGS (cMAGs) assembled from five human fecal samples, sequenced with PacBio HiFi technology. While the samples and methodologies used are not novel, the workflow provided by the authors to detect and filter out biased cMAGS is interesting, combining multiple quality control methodologies from marker genes detection to GC bias exploration.

As the actual state of the paper is providing an interesting workflow but missing clarity in the pipeline and closely related work that used the same samples, I think that the paper should be accepted after incorporating the following suggestions:

Major comments:

1. Most samples used in this study were already used to build MAGs by Feng et al (<https://doi.org/10.1038/s41592-022-01478-3>). Consequently, this study provides a resource with limited novelty. Comparison with MAGs published by Feng et al. would be welcomed.

As the reviewer mentioned, both of the Feng et al. study and ours used the HiFi metagenomic sequencing data from four samples (PacBio Data Release) and the same the assembly tools for HiFi sequencing data. However, our study has significant novelty compared to Feng et al in terms of (1) the purpose of the study, (2) method development, and (3) our new HiFi metagenome sequencing data resources.

(1) The purpose of the study:

The main research purpose of the Feng et al. was development of an improved assembler for HiFi long-read sequencing. They mentioned ‘near complete MAGs’ while comparing performance of their assembler, hifiasm_meta, and others, because high completeness of MAG is the indication of an improved assembly performance. In the Feng et al. paper, the authors discuss about near-complete contigs/MAGs (by CheckM completeness > 90% and contamination <5%) but never mentioned complete circular MAG (cMAG).

In contrast, our study focused on circular contigs and cMAGs which did not go through binning process. For the first time, to the best of our knowledge, our study proposed a method for distinguishing cMAGs from defective circular contigs by utilizing congruence with conspecific genomes, and eventually released 102 circular and complete prokaryotic genomes for diverse taxa. Furthermore, we investigate the properties of cMAGs by HiFi sequencing such as nucleotide accuracy, taxonomy diversity and ability in large genome assembly.

(2) Method development

Feng et al. developed a new assembly tool for HiFi sequencing data, hifiasm_meta, and compared it with other long-read sequence assembly tools such as metaFlye and HiCanu. Although assembly with hifiasm_meta (and other assemblers) generates many circular contigs, not all of them represent circular complete prokaryotic genomes. Those circular contigs also include plasmid, virus, and faulty circularized

bacterial DNA. Therefore, we developed a computational workflow and a software cMACfilter for distinguishing complete prokaryotic genomes from many of circular contigs by long-read assemblers such as hifiasm meta.

(3) New data resources

Feng et al. used only public data. Our study not only use the public data but also presents new HiFi metagenomic sequencing data from a Korean fecal sample (KR001) with ultra-depth (~30 Gb). In fact, the new sequencing sample was the first HiFi metagenomics sequencing data for Korean gut microbiome. While the public data were based on pooled samples, our data were derived from a single individual sample. Thus, the new sequencing sample allowed us to assess single-sample assembly of HiFi sequencing read. In addition, its ultra-deep sequencing resulted in cMAGs with genome size that is larger than 99% of reference human gut bacterial species (>6Mbp). In particular, *Bacteroides* genus was previously reported as the taxa that show more diversity in non-western populations (Genome Medicine 13:134, 2021; <https://doi.org/10.1186/s13073-021-00950-7>). Therefore, KR001 sample provides cMAGs that are under-sampled in the western population. To highlight significant addition of cMAGs by KR001 sample, we have added the following text in the revised manuscript.

“We found that 29 of 102 cMAGs were exclusively assembled from the newly sequenced Korean fecal sample (KR001) (Fig. 1c).”

“Among them, 1 phylum, 1 class, 1 order, 4 families, 12 genera, and 22 families are contributed solely by the KR001 sample (Fig. 1d-e, Supplementary Table S5-6).”

In addition, our study provides 102 cMAGs covering diverse taxa: 2 domains, 9 phyla, 11 classes, 14 orders, 24 families, 52 genera, and 84 species. Notably, 39 cMAGs belong to uncultured species (that is, there was no isolated genome for the species). All these cMAGs will be valuable resources for the study of human gut microbiome.

2. The workflow created by the authors is interesting, but raises multiple questions. For example, if I understand correctly, one step of the filtering remove all contigs with less than 5 conspecific HRGM genomes and with less than 95% core contig retrieval. These parameters seem penalizing for under-represented genomes, or genomes that are highly distributed in the human population that include geographic microbial disparities. This will for instance favor species with lower diversity, such as the cMAG used as validation GCF_000224965, which is in fact a probiotic bacterium widely used in food industry (see DOI: 10.1128/JB.06079-11).

This is an insightful comment. We agree that our workflow could miss the potential cMAGs for novel or under-represented species genomes. Indeed, two circular contigs (VEG02_HAM_0059, VEG01_HIC_0059) that passed our first filtration step were excluded from the final cMAG list because they do not have sufficient conspecific genomes for testing congruence (0, and 1 respectively) in the HRGM catalog.

These circular contigs could be either novel species genome or defective ones. Given that the number of metagenomic samples used for the construction of HRGM

(>12,000 samples) is much larger than the number of samples used in this study (5 samples), finding a novel species genome is highly unlikely. Then, the circular contigs with no conspecific genomes are more likely to be defective genome. In addition, both circular contigs have the same GTDB-Tk warning "*Genome not assigned to closest species as it falls outside its pre-defined ANI radius*". It means that these circular contigs do not have their conspecific genomes even in the GTDB database, which contains much larger number of prokaryotic genomes. This also strengthens our reasoning that the chance of being novel species genome is quite low. But we agree that the result was only for the two circular contigs and there is still a possibility to miss some under-represented genomes with our method.

Nevertheless, in this study, we are committed to develop a robust method for selecting complete circular MAGs from many circular contigs and investigate their characteristics. We conducted analyses with weighting more on specificity than sensitivity. Therefore, we decided to exclude circular contigs with insufficient number of conspecific genomes from our final cMAG list.

We also agree that cMAGfilter could be favorable for species with lower diversity. Achieving a high retrieval rate will be easier with species with low-diversity compared to that with high-diversity. Therefore, applying the same analytical parameters for every species actually requires a different level of intactness for different species genomes. Although it is possible to optimize the parameters for different species, it will require a really extensive investigation and possess many risks (e.g., applying a wrong parameters for under-represented taxa). Notably, we could assemble cMAGs for both of a species with extremely low-diversity (*Bifidobacterium animalis*, Fig. 2a) and a species with extremely high-diversity (*Collinsella aerofaciens* and their close neighbors; the most dispersed branch of the phylogenetic tree, Fig. 1d; Genome Medicine 13:134, 2021, doi: 10.1186/s13073-021-00950-7). This result suggests that our parameters reasonably work well for both low- and high-diversity species.

Nevertheless, we agree that the reviewer's points. Thus we discussed these limitations in the discussion section as follows.

"There are limitations in our study. Our method for selecting cMAGs relies on the congruence with their conspecific genomes, but not all cMAGs have sufficient number of conspecific genomes for evaluation (e.g., ≥ 5 in this study). This requirement may penalize under-represented genomes and potentially result in the exclusion of novel species genomes. Indeed, we had to exclude two circular contigs that could not be evaluated for this reason. In addition, we tested only five HiFi metagenomic sequencing samples in this study. Therefore, some of the results (e.g., performance of each assembler) and the cMAGfilter parameters may not be generalized."

The authors should then discuss the necessity to use core contigs, ie contigs >5kbp long that are similar in 80% of conspecific genomes, instead of the core genes only.

We agree with the reviewer's point that we could develop the method using the core genes rather than core contigs. Because the genome congruence measure based on

core genes has the same rationale, we expect the result will be very similar. However, the gene-based method requires an additional and significant computational cost for gene calling. Also, comparison with nucleotide sequences of core contig is more intuitive than comparison with protein sequences of core genes. We have added this discussion in the revised manuscript as follows.

“We assessed completeness of circular contigs using core contigs rather than core genes share among conspecific genomes. Conceptually, we may use core genes for the analysis of congruence of circular contigs with their conspecific genomes instead. However, gene-based methods need to conduct an additional process for gene calling from the circular contigs which requires high computational cost. Moreover, comparison with protein sequences of the core genes would be less intuitive than comparison with nucleotide sequences of core contigs. Thus, the contig-based congruence analysis provides computational advantages over the gene-based analysis.”

The minimum length threshold of the contig is to exclude less reliable contigs (the shorter contigs tend to be more vulnerable in the binning process) and the threshold for the percentage of the conspecific genome is to find highly-shared contigs. To improve the clarity, we have added these additional description in the Methods section as follow.

“A core contig was defined as longer than 5 kbp (to exclude short sequences which tend to be more vulnerable) and present in more than 80% of conspecific genomes (to select highly shared contigs).”

We determined these parameters based on our benchmark with the cMAGs with conspecific reference-quality genomes. We agree that the number of samples in this study was not sufficient enough to set the optimal parameters. The optimal parameters can be affected by taxonomy, sequencing depth, and complexity of the metagenome. We developed the tool flexibly. Users can tune their cMAGfilter parameters.

Also, are the samples pooled prior to assembly? It would be nice to make this more transparent in the Fig 1.

The four samples from the PacBio data release were pooled before sequencing (at the sample preparation step). Our in-house sample (KR001) is not a pooled sample. The fecal sample was obtained from a Korean individual. As the reviewer suggested, we included this information in Figure 1 legend and modified the Methods section as follows.

“OMN and VEG represent the public pooled samples derived from the omnivores and vegans, respectively, and KR001 represent the newly sequenced Korean sample.”

“The samples were collected from US and pooled before the sequencing.”

3. 7 complete MAGs were recovered with HiCanu only. Contrary to the two others assemblers, HiCanu was not designed to process metagenomic data. Could the authors explain why these CMAGs were missed by Hifiasm-meta and metaflye?

Although the HiCanu is not designed for metagenomic assembly, we used the parameters recommended for metagenomic assembly by the developers (<https://canu.readthedocs.io/en/latest/faq.html?highlight=batMemory#what-parameters-can-i-tweak>). We also investigated whether the 7 cMAGs that have been assembled with HiCanu only have some sign of defectiveness. However, all the quality measures were not significantly different from the other cMAGs (See figure below). We also did not find any distinguishing properties of the 7 cMAGs. We guess that the difference in algorithm may result in difference in sensitivity. However, we believe, in-depth algorithmic comparison between different assemblers might be out of scope of our study.

4. “Genomic island proteins are more likely to be discovered from cultured taxa with isolated genomes.” “Thus, we hypothesized that cMAGs would provide opportunities to discover novel genome island genes.” These statements are probably wrong. If a region is missing in a MAG, it does not mean it could not be assembled. Contigs corresponding genomic island are present in metagenomic assemblies obtained with short-reads. Yet, they are missing in MAGs because binning tools often miss them.

We agree with the reviewer’s comment. Probably we do not see GI in the MAGs, because genomic regions for mobile genetic elements were assembled but could not binned into a MAG. Then, assembled contigs containing GI genes may exist and can provide information about GI genes. Therefore, as reviewer point out, cMAG does not make significant contribution in discovery of novel GI genes.

Nevertheless, we still can hypothesize that GI genes in the complete or near complete genomes could be more likely to be functionally characterized. Indeed, we could observe higher functional annotation (eggNOG ortholog) for GI genes of cultured taxa (with isolated genomes) than uncultured taxa (Fig. 4e). This suggest that similar to isolated genomes for cultured taxa, cMAGs will facilitate functional characterization of GI genes. We have modified the text in the revised manuscript as follow.

“...genomic island regions seem to be the leading cause of gaps in MAGs obtained by short-read sequencing. We hypothesized that genomic island proteins could be more likely to be functionally characterized from cultured taxa with isolated genomes. Indeed, genome island proteins of cultured taxa showed a higher eggNOG ortholog²⁰ annotation rate than uncultured taxa (Fig. 4e). This suggests that cMAGs would provide better

opportunities to investigate uncharacterized genome island genes.”

5. Similar studies previously published are not cited but cover the same topic.

We appreciate the reviewer’s suggestions. We could not cite some very recent papers, because the metagenomic assembly is a fast moving research field. We have added following references in the introduction of the revised manuscript.

<https://doi.org/10.1038/s41592-022-01478-3>

This paper was already cited as a reference for hifiasm_meta assembler (Ref. 11). We did not cite this paper regarding complete MAGs, because it presents near complete MAGs but not complete MAGs.

<https://doi.org/10.1080/19490976.2021.2021790>

“PacBio SMRT long-read sequencing with ultra-deep short-read sequencing also assembled four cMAGs from 12 human fecal samples but their nucleotide accuracy was not reported (Ref. 7).”

<https://doi.org/10.1128/mra.00250-22>

This is a genome announcement paper about a public release of nine cMAGs from HiFi metagenomics sequencing. This paper was published while our manuscript was under review process. Our manuscript was uploaded at preprint server on February 10 (<https://www.biorxiv.org/content/10.1101/2022.02.09.479829v1>) and the manuscript for this paper was received by Journal on March 10 and published on May 9. Therefore, we included this reference in Discussion section of the revised manuscript as follows.

“While this manuscript is being peer reviewed, a public release of nine cMAGs assembled by HiFi sequencing was announced by independent research group (Ref. 26). However, the study did not present their systematic evaluations on nucleotide accuracy, taxonomic diversity, and ability in large genome assembly, probably due to the insufficient number of retrieved cMAGs.”

Minor comments:

1. 1. 21 : replace “discontinuity” by “fragmentation”

Corrected as suggested.

2. 1. 26 : replace “were filtered » by “were selected”

Corrected as suggested.

3.1.26, 79: “authentic prokaryotic genomes” this term is too vague. Please rephrase.

First of all, we apologize for this confusing word ‘authentic’ in the text. We originally used the word to stress that we excluded any circular contigs that did not derived

from prokaryotic genomes (e.g., plasmids, viral genomes, and etc.). But, we realized that ‘complete prokaryotic genomes’ is sufficient to deliver our meaning. Thus, we have replaced the words ‘authentic prokaryotic genomes’ with ‘complete prokaryotic genomes’ in the revised manuscript.

4. 1. 29,30 : “whose genomes have not been characterized yet.” This is not true: <https://doi.org/10.1186/s12864-020-06807-4>

Thanks for the comments. We checked the paper and noticed that the authors characterized metabolic pathways of RF39 lineage species using MAGs. Although characterization of genomes for TANB77 order has not been reported yet, we have removed the sentence in the revised manuscript to avoid confusions.

5. 1. 32 : “of human gut microbiota” This may be true for other microbial communities

We agree with the reviewer’s comment. We have modified the sentence as follows.

“HiFi metagenomic sequencing will facilitate cataloging accurate and complete genomes from complex microbial communities, including uncultured species.”

6. 1.35: replace “culturomics technology” by “culturomics techniques”

Corrected as suggested.

7. 1. 41: genome fragmentation is a consequence of “conserved, repetitive, and mobile sequences”, but not chimerism

We have removed “chimerism” as suggested.

8. 1. 49 : HiCanu was not designed for metagenomic assembly. Please cite MetaFlye. The name of the assemblers should be explicitly written

We have modified the sentence as suggested.

“Moreover, specialized assemblers for HiFi metagenomic assembly, hifiasm-meta (11) and metaFlye (13), enable the highly accurate reconstruction of cMAGs.”

9. The end of the introduction is the abstract the paper.

We have removed redundant text at the end of introduction and rewrite the last paragraph of the introduction as follows.

“In the present study, we conducted an exhaustive assembly of HiFi metagenomic sequencing reads from five human fecal samples. Seeking only circularized contigs, we skipped the binning procedure. We developed a bioinformatics workflow to filter initially assembled circular contigs for complete prokaryotic genomes. Eventually, we obtained 102 cMAGs and investigated their nucleotide accuracy, taxonomic diversity, and ability in large

genome assembly. We also examined how HiFi metagenome sequencing can complement short-read metagenome sequencing for the study of uncultured species in the microbial communities.”

10. 1.67 & 1.277 The URL does not work. The authors should provide a resource available on the INSDC

We have removed the URL because it gets obsolete. Instead, we have provided the NCBI-SRA accession code in Data Availability section.

“The public HiFi metagenomic sequencing data for pooled human fecal samples are available from NCBI Sequence Read Archive (PRJNA750084). HiFi metagenomic sequencing data for a Korean fecal sample generated in this study are deposited in the Sequence Read Archive (PRJNA798244).”

11. L119 : “Although bias toward a few bacterial clades (e.g., Oscillospirales) was observed” Is this bias related to the estimated taxonomic composition of samples?

We observed a significant positive correlation between the number of obtained cMAGs and taxonomic abundance at both family and order levels (See figure below). Thus, the observed bias might be in partial based on taxonomic composition. To incorporate this result, we have modified the sentence as follows.

“Although bias toward a few abundant bacterial clades (e.g., Oscillospirales) was observed,...”

We also observed some exceptions. Lachnospirales was the most abundant order but the number of assembled cMAGs (8) was much fewer than that for Oscillospirales (34). This suggests that other factors (e.g., genome size and degree of repetition) could also affect the retrieval of cMAGs by HiFi sequencing and assembly.

12. L126: “same-strains from different samples are hardly identical to each other because of genetic variation accrued via horizontal gene transfer and spontaneous mutation.” I don’t understand this sentence. Same strains are supposed to be identical. Please rephrase.

The “same-strains” was replaced with “same-species”.

13. L129: The chosen CMAG probably derives from a probiotic strain which explains the very high similarity with GCF_000224965.2. I would mention it explicitly.

We have added the information in the revised manuscript as follows.

“Given that the RefSeq genome (GCF_000224965.2) is for a probiotic bacterium, B. animalis subsp. lactis BLC1 (Ref. 17), the cMAG (OMN01_MFL_0491) may also be derived from probiotics.”

14. L. 174 This is not true. Plaza Onate et al. (doi: 10.1128/mra.00250-22) published complete MAGs for these two orders.

The resource announcement paper by Plaza Onate et al. was received on March 10 and published on May 9. Our manuscript was submitted and uploaded on preprint server on February 10. Therefore, Plaza Onate et al. paper was submitted no earlier than ours but rapidly reviewed and published while our manuscript was still under peer review process. Thus, our statement was correct at the time of our initial submission but not for now as reviewer pointed out. Therefore, we have removed the words “for the first time” in the revised manuscript.

15. L. 193: Please, provide a reference (<https://doi.org/10.1099/mgen.0.000436> for instance)

We have added the reference (Ref. 5).

16. L199 The term “genome coverage” is ambiguous and might be confounded with sequencing coverage. Please define the terms more precisely.

We understand the reviewer’s concern that the term "genome coverage" may confuse readers. Therefore, we replaced the term with "retrieval rate" (more explicitly genome bin retrieval rate by its conspecific genomes), and also described what the term means and add reference to the corresponding method section that has clear explanation of the terminology.

17. L.215 “ONT metagenomic sequencing samples did not reach the median genome island length” ONT can achieve longer read length by changing some parameters when building sequencing libraries. I don’t think this statement is relevant

We have removed the sentence as suggested.

18. L228 : metaFlye should perform well with HiFi metagenomic sequencing data. However, it might not handle pooled samples that contain multiple closely related strains as well as hifi asm meta.

Because we have added a new single-sample HiFi sequencing data (KR001) for this study, we could compare retrieval of cMAGs from the single sample assembly between the three different assemblers. We found that metaFlye assembled the least number of cMAGs from the KR001 sample (See Figure below). Although, we may not generalize the result based on this single study, it is unlikely that sample pooling was the main factor that causes the lower retrieval of cMAGs by metaFlye.

19. L242 This issue is addressed in checkm v2. The authors could mention it.

We appreciate the reviewer’s comments. We found that the low completeness of the uncultured clades (TANB77 and RF39) are significantly improved by CheckM2 (See figure below). We have added the CheckM2 result of cMAGs in the Supplementary Fig.7 and Supplementary Table. We have also added the following text in the revised manuscript.

“Notably, a recent update of the CheckM2 also addressed this issue (Ref. 24). Indeed, we observed increased completeness of TANB77 cMAGs with CheckM2 (Supplementary Fig. 7, Supplementary Table S7)”

20. L271 It is not a matter of abundance (i.e concentration) but a matter of quantity (i.e mass).

We have replaced the word ‘abundant’ with ‘high molecular weight DNA’.

21. L276: replace “datasets” by “samples”

Corrected as suggested.

22. Fig1B Replace the Venn Diagram with an upsetR figure

We have revised the figure as suggested.

Number of cMAGs by assemblers

Thank you again for your time and effort, and for helping to improve the manuscript. We hope that these changes have made it more appropriated for publication, and we look forward to your response.

REVIEWERS' COMMENTS

Reviewer #1 (Remarks to the Author):

This review thanks the authors for revising their manuscript which improved from its original version.

Reviewer #2 (Remarks to the Author):

The precise and detailed answers provided by the authors are satisfactory and addressed my main concerns.

Overall, the revised manuscript is ready for publication. Yet, I have still a few questions and remarks.

1. In the last section about genomic islands (GI), the authors claim that they are missing in most MAGs because they are hard to assembled using short reads. Although this statement is true, Maguire and colleagues (doi: 10.1099/mgen.0.000436) also show that GIs may missed because binning tools fail to recover them.

Thus, I would suggest to modify some sentences accordingly. For instance:

“hard-to-assemble” -> “hard-to-recover”

“[GIs] are notoriously difficult to retrieve from short-read metagenomic assembly” -> “[GIs] are notoriously difficult to retrieve from short-read metagenomic assembly and to group into MAGs during the binning step.”

2. I don't understand a new hypothesis that has been added in the revised manuscript.

“Indeed, genome island proteins of cultured taxa showed a higher eggNOG ortholog annotation rate than uncultured taxa (Fig. 4e).

Indeed, GIs from cultured taxa showed higher annotation rate because they are better characterized and more studied in the laboratory. I do not understand the reasoning that leads the authors to claim that:

“This suggests that cMAGs would provide better opportunities to investigate uncharacterized genome island genes.”

3. Please specify in the title and/or in the abstract that the HiFi technology is developed by the company Pacific Biosciences (PacBio)

4. I don't think the term "isolated genomes" is appropriate. Please replace it with a more precise term like "genomes derived from isolates"

Reviewer #2 (Remarks to the Author):

The precise and detailed answers provided by the authors are satisfactory and addressed my main concerns.

Overall, the revised manuscript is ready for publication. Yet, I have still a few questions and remarks.

1. In the last section about genomic islands (GI), the authors claim that they are missing in most MAGs because they are hard to assembled using short reads. Although this statement is true, Maguire and colleagues (doi: 10.1099/mgen.0.000436) also show that GIs may missed because binning tools fail to recover them.

Thus, I would suggest to modify some sentences accordingly. For instance:

“hard-to-assemble” -> “hard-to-recover”

“[GIs] are notoriously difficult to retrieve from short-read metagenomic assembly” -> “[GIs] are notoriously difficult to retrieve from short-read metagenomic assembly and to group into MAGs during the binning step.”

We appreciate with your comment, we totally agree and changed as suggested.

2. I don't understand a new hypothesis that has been added in the revised manuscript.

“Indeed, genome island proteins of cultured taxa showed a higher eggNOG ortholog annotation rate than uncultured taxa (Fig. 4e).

Indeed, GIs from cultured taxa showed higher annotation rate because they are better characterized and more studied in the laboratory. I do not understand the reasoning that leads the authors to claim that:

“This suggests that cMAGs would provide better opportunities to investigate uncharacterized genome island genes.”

Because the genome island region of uncultured taxa are difficult to be retrieved from short-read based

Thanks for your comments, firstly, we want to explain the reason why we wrote the sentence.

- 1. The less studied genome island regions of the uncultured taxa cannot be assembled from the sequencing of isolated microbes; therefore, they can only be studied from metagenomic sequencing.**
- 2. Genome island regions are difficult to bin properly by short-read based MAG assembly pipeline (as shown in Fig4a, d); therefore, they can be better identified by long-read sequencing.**

However, we understand your concern that although GI regions are hard to bin, they still can be assembled by short-read sequencing. Also, genes from un-binned contigs can be functionally characterized.

Therefore, we removed the sentence from the manuscript.

3. Please specify in the title and/or in the abstract that the HiFi technology is developed by the company Pacific Biosciences (PacBio)

We added the company name on the Abstract section.

4. I don't think the term "isolated genomes" is appropriate. Please replace it with a more precise term like "genomes derived from isolates"

We changed the terminology as suggested.